# Bias and variance of the Bayesian-mean decoder

**Arthur Prat-Carrabin**
Department of Economics
Columbia University
New York, USA
`arthur.p@columbia.edu`

**Michael Woodford**
Department of Economics
Columbia University
New York, USA
`mw2230@columbia.edu`

## Abstract

Perception, in theoretical neuroscience, has been modeled as the encoding of external stimuli into internal signals, which are then decoded. The Bayesian mean is an important decoder, as it is optimal for purposes of both estimation and discrimination. We present widely-applicable approximations to the bias and to the variance of the Bayesian mean, obtained under the minimal and biologically-relevant assumption that the encoding results from a series of independent, though not necessarily identically-distributed, signals. Simulations substantiate the accuracy of our approximations in the small-noise regime. The bias of the Bayesian mean comprises two components: one driven by the prior, and one driven by the precision of the encoding. If the encoding is 'efficient', the two components have opposite effects; their relative strengths are determined by the objective that the encoding optimizes. The experimental literature on perception reports both 'Bayesian' biases directed towards prior expectations, and opposite, 'anti-Bayesian' biases. We show that different tasks are indeed predicted to yield such contradictory biases, under a consistently-optimal encoding-decoding model. Moreover, we recover Wei and Stocker's "law of human perception" [1], a relation between the bias of the Bayesian mean and the derivative of its variance, and show how the coefficient of proportionality in this law depends on the task at hand. Our results provide a parsimonious theory of optimal perception under constraints, in which encoding and decoding are adapted both to the prior and to the task faced by the observer.

## 1 Introduction

Perception has been described in neuroscience as resulting from a two-stage, 'encoding-decoding' process. In the encoding stage, an external stimulus elicits in the brain an internal representation (in the form of a neural activity) whose statistics depend on the physical properties of the stimulus. In the decoding stage, the brain makes use of the internal representation to achieve a goal: for instance, to estimate the magnitude of the stimulus, or to choose the best of two stimuli. It is generally understood that the encoding capacity of the brain is limited, i.e., that there is some amount of imprecision in the internal representation. The theory of 'efficient coding', however, proposes that the encoding is optimized, under a constraint on the encoding capacity of the brain; and the theory of the 'Bayesian brain' puts forward the notion that the brain optimally decodes the imprecise representation, through Bayesian inference. Although these two theories emerged separately, recent models of perception combine the two into a single, normative account of perception [2–9].

There does not seem to be a general consensus, however, as to what objectives the encoding and decoding stages optimize [10]; in some models (though not all [8]), the two stages optimize different objectives [6, 9]. Furthermore, in behavioral experiments involving one-dimensional, physical magnitudes (such as some orientation or length), human perceptual decisions are often biased, and they exhibit variability ('noise'). Recent encoding-decoding models of perception account for these

35th Conference on Neural Information Processing Systems (NeurIPS 2021).

patterns, and make specific predictions regarding how, in perceptual tasks, the bias and the variance of the estimates should relate to the distribution of stimuli (the prior). They predict, in addition, that the bias should be proportional to the derivative of the variance. Although there is substantial support for this relation in experimental data [1], the value of the constant of proportionality — including its sign — remains unclear; it is thus uncertain, in a given task, whether the perceptual bias should be in one direction, or the other. Experimentally, biases towards the stimuli that are more probable as per the prior [11–14], and the opposite, i.e., biases towards the less probable stimuli ('anti-Bayesian percepts') [6, 15–17], have both been exhibited. Brayanov and Smith [17] simultaneously find the former in a task involving (implicit) weight estimation, and the latter in a task involving weight discrimination.

The bias, in encoding-decoding models of perception, is in great part determined by the 'decoder', i.e., the decoding mechanism. A widely used decoder is the Bayesian mean. The Bayesian mean is the optimal decoder both in estimation tasks (with squared-error loss), and in a common type of discrimination tasks (which we describe in more detail below). It is, in the general case, biased; its bias depends on the prior, but also on the characteristics of the encoding. Here, we present an analytical approximation to the bias of the Bayesian mean, obtained under very general assumptions about the encoding. In short, the main assumption is that the encoding takes the form of a series of independent signals. These signals need not be identically distributed. This corresponds, for instance, to a population of sensory neurons that are each characterized by a different tuning curve. We do not endeavor to model the individual properties of each neuron in a population: instead, we rely on the statistical properties resulting from the accumulation of independent signals, as summarized by their total Fisher information. This approach also provides an approximation to the variance of the Bayesian mean. A mathematical result similar to that presented here has been derived in the statistics literature [18], but to our knowledge its importance for perception models has not been noted.

Two terms drive the bias of the Bayesian mean: the variations of the prior across stimulus space, and the variations of the Fisher information. When the encoding is 'efficient', these two terms have opposite effects. The resulting size — and sign — of the bias will depend on the specifics of the encoding, and on the objective it optimizes. We show, below, how the encoding objective function determines the relation between the bias and the prior, the direction of the bias, and the relation of proportionality between the bias and the derivative of the variance. Our results explain why opposite biases are obtained in different experiments: when the encoding is optimized for an estimation task (with squared-error loss), the bias is directed towards the more probable stimuli; while when the encoding is optimized for discrimination tasks, the opposite is predicted. Finally, we run simulations, and examine how the accuracy of our asymptotic approximations vary with the amount of imprecision in the encoding.

## 2    Bias and variance of the Bayesian mean

We consider an observer presented with a stimulus characterized by a magnitude, $x$ (for instance, its length, its orientation, or its speed), which is randomly sampled from a prior distribution, $\pi(x)$. We assume that the presentation of the stimulus elicits in the brain of the observer an internal representation, $r$, as a series of $n$ random signals, $r_1, \ldots, r_n$, sampled independently from $n$ distributions conditioned on the stimulus, $f_i(r_i|x)$. We assume in addition that moments of a sufficient order exist for all $i$, and that the rate of growth of these moments is limited in a way that allows us to use a central limit theorem (see Supplementary Materials). This is the encoding stage. We emphasize that we assume the signals to be independent, but not necessarily that they are identically distributed. They can be understood as representing the activity of a neural population in which each neuron has a different tuning curve, for instance centered on a 'preferred' stimulus. (The result we present is however a mathematical one, and is thus not tied to a particular physical implementation.) The internal representation, $r$, is characterized by its Fisher information, $I(x)$, which is the sum of the Fisher informations of each signal; it is thus of order $n$. Our calculations rely on the asymptotic statistical properties resulting from the accumulation of these signals, when $n$ is large. The Fisher information plays here a central role, by providing a summary measure of the sensitivity of the representation to the stimulus, effectively abstracting away the specifics of each random signal (such as, for instance, the individual properties of each neuron in a population).

Turning to the decoding stage, we assume that our observer uses the Bayesian mean. We sketch, here, the derivation of the approximations to the bias and to the variance of this decoder (a more

detailed derivation is presented in the Supplementary Materials). Given an internal representation, $r$, the Bayesian mean is

$$\hat{x}(r) \equiv \mathbb{E}[x|r] = \frac{\int \tilde{x}\pi(\tilde{x})\prod_{i=1}^{n} f_i(r_i|\tilde{x})\,\mathrm{d}\tilde{x}}{\int \pi(\tilde{x})\prod_{i=1}^{n} f_i(r_i|\tilde{x})\,\mathrm{d}\tilde{x}}. \tag{1}$$

Fixing $r$, we denote by $L(x)$ the log-likelihood of a stimulus $x$, i.e., $L(x) = \sum_{i=1}^{n} \ln f_i(r_i|x)$. The log-likelihood is thus of order $n$. We rewrite Eq. (1) using the log-likelihood, as

$$\hat{x}(r) = \frac{\int \tilde{x}\pi(\tilde{x})e^{L(\tilde{x})}\,\mathrm{d}\tilde{x}}{\int \pi(\tilde{x})e^{L(\tilde{x})}\,\mathrm{d}\tilde{x}}. \tag{2}$$

This equation is the ratio of two integrals of the type $\int g(\tilde{x})e^{L(\tilde{x})}\,\mathrm{d}\tilde{x}$. With large $n$, the integrands are dominated by their values at the point where $L(x)$ reaches its maximum, i.e., at the maximum-likelihood estimator (MLE). Through Laplace approximations of these integrals, we obtain an approximate expression of the Bayesian mean, equal to the MLE corrected by a term of order $1/n$ that depends on the prior, on the likelihood, and on their derivatives. The MLE has been widely studied in statistics: approximations to its bias and to its variance, as functions of the Fisher information, $I(x)$, are well known [19–21]. This allows us to obtain an approximation of order $1/n$ to the bias of the Bayesian mean, as

$$\mathbb{E}\big[\hat{x} - x \,\big|\, x\big] \simeq b(x) = \frac{1}{I(x)}\left[\frac{\pi'(x)}{\pi(x)} - \frac{I'(x)}{I(x)}\right], \tag{3}$$

where $x$ is the true value of the stimulus, and $\pi'$ and $I'$ are the derivatives of the prior and of the Fisher information. We emphasize that this is a very general result about the Bayesian mean, obtained under one main assumption, that a series of independent signals, not necessarily identically distributed, encodes the stimulus.

Equation (3) shows, first, that the bias of the Bayesian mean is scaled by the inverse of the Fisher information, $1/I(x)$. In other words, the more precise the internal representation about the stimulus, the smaller the bias. Second, the bias depends on the difference between the relative variation of the prior (around $x$), $\pi'(x)/\pi(x)$, and the relative variation of the Fisher information, $I'(x)/I(x)$. The former induces a 'prior effect', by which the Bayesian mean tends to be biased towards more probable stimuli. The latter induces an 'encoding effect', by which the bias tends to be in the direction of decreasing precision of the encoding. The resulting direction of the bias is the sum of these two effects. If more probable stimuli are encoded with a higher precision (i.e., if $\pi'$ and $I'$ have the same sign), which is what efficient-coding models suggest (see below), then these two effects are opposite. If in addition the encoding effect is stronger than the prior effect, then the Bayesian mean is biased towards stimuli of *decreasing* probability, a bias direction that has been qualified as 'anti-Bayesian' [6, 17], although here it is obtained with a Bayesian-mean decoder.

Finally, it is well known that the variance of the MLE converges to the inverse of the Fisher information, $1/I(x)$. It is thus the leading term of the variance of the Bayesian mean, and we find that the additional terms, in the approximation to this variance, are of orders higher than $1/n$. Hence the inverse of the Fisher information is our approximation of order $1/n$ to the variance of the Bayesian mean, i.e.,

$$\mathrm{Var}\big[\hat{x} \,\big|\, x\big] \simeq v(x) = \frac{1}{I(x)}. \tag{4}$$

The Bayesian-mean decoder is thus less variable for stimuli that are encoded with higher precision. Equations (3) and (4) underline how the bias and the variance of the Bayesian mean depend on the precision of the encoding, as measured by the Fisher information. Different choices of encoding thus result in different behaviors of the bias and of the variance. We now turn to the question of how the choice of encoding can be efficiently adapted to the specific task faced by the observer.

## 3  Efficient coding

Perception allows obtaining some information about the environment, which can then be used to make decisions. In many situations, an increased precision about the relevant variable (here, the stimulus $x$) results in better decisions, and higher rewards. In our setting, the Fisher information of the encoding stage is a measure of this precision. With an unbounded Fisher information ($I(x) \to \infty$), i.e., with

infinite encoding precision, the variable is known exactly: both the bias and the variance vanish (Eqs. (3) and (4)). It is however unlikely that the brain can reach infinite precision in the encoding stage, as both the number of neurons and their firing rates are bounded quantities. Efficient-coding models, accordingly, assume that there is a limit to the brain's encoding capacity. Within this limit, however, the encoding is assumed to be chosen so as to optimize some objective. Below, we derive the optimal encoding for each of several objective functions, and we examine the resulting bias and variance of the Bayesian-mean decoder.

## 3.1 Encoding objectives for different tasks

As the Bayesian-mean decoder is the one that minimizes the expected squared error, we start by examining the implications for the encoding stage of this objective function (which corresponds to an estimation task). Given a stimulus, $x$, the expected squared error, averaged over the internal representations, $r$, is approximately the inverse of the Fisher information, $1/I(x)$ (see Supplementary Materials). Given a prior distribution of stimuli, $\pi(x)$, the expected square error, averaged over the stimuli, $x$, is thus approximately

$$\int \frac{\pi(x)}{I(x)} \, \mathrm{d}x. \tag{5}$$

An encoding efficiently adapted to this estimation task is thus one in which the Fisher information minimizes this objective function, under a constraint.

Before further specifying this optimization problem, we examine other objectives, that are implied by other tasks. An interesting task is one in which the observer is asked to choose between two presented stimuli, $x_0$ and $x_1$, and then receives a reward equal to (or proportional to) her choice, i.e., the observer "gets what she chooses" [22], which is generally the outcome in many choice situations (we call this a "discrimination task with proportional rewards": it is different from a "discrimination task with constant rewards", in which the observer obtains a constant reward if the chosen stimulus is the correct one, and zero otherwise [23]; we consider the latter case further below). With this task, assuming that the two stimuli elicit in the brain of the observer two independent internal representations, $r_0$ and $r_1$, the optimal decoding strategy is to estimate and compare the expected rewards implied by either choice, i.e., to compute the expected values of the two stimuli, $\hat{x}(r_0)$ and $\hat{x}(r_1)$. The Bayesian mean, thus, is also the optimal decoder for this task. This calls for the examination of the optimal encoding in this case. Given two stimuli, $x_0$ and $x_1$, the observer is guaranteed to obtain at least the smaller of the two, thus her loss if she chooses the wrong one is the absolute difference between the two, $|x_1 - x_0|$. Hence she aims at minimizing the expected loss

$$\iint P(\mathrm{error}|x_0, x_1)|x_1 - x_0|\pi(x_0)\pi(x_1) \, \mathrm{d}x_0 \, \mathrm{d}x_1, \tag{6}$$

where $P(\mathrm{error}|x_0, x_1)$ is the probability of choosing the wrong stimulus (the smaller one); this probability depends on the Fisher information that characterizes the observer's encoding stage, $I(x)$. We find (see Supplementary Materials) that with large $n$ the expected loss is approximated by the quantity

$$\int \frac{\pi^2(x)}{I(x)} \, \mathrm{d}x. \tag{7}$$

In both this discrimination task and the estimation task above, a larger Fisher information results in a smaller loss, and thus better decisions. However, the objective of this discrimination task (Eq. (7)) and that of the estimation task (Eq. (5)) are not the same. This yields qualitative differences in perception, which we examine further below.

The two encoding objectives presented so far are such that the decoding used by our observer (the Bayesian mean) optimizes these objectives, resulting in an encoding-decoding mechanism consistently optimized for these objectives. Another encoding objective studied in the literature is the maximization of the mutual information between the stimulus and the internal representation, $MI(x, r)$. This objective has been seen as a 'general-purpose' encoding objective, that is useful for many different tasks, although not optimal for most [6, 10]. As shown by Clarke and Barron [24], when the internal representation is a series of $n$ signals (as we have assumed), the mutual information can be approximated using the Fisher information, as

$$MI(x, r) = \frac{1}{2} \ln \frac{n}{2\pi e} + H(X) + \frac{1}{2} \int \pi(x) \ln I(x) \, \mathrm{d}x + o(1), \tag{8}$$

where $H(X)$ is the entropy of the stimulus. The first two terms of this equation do not depend on the Fisher information. As for the third term, we follow Ref. [9] in noting that $\ln I(x) = \lim_{\alpha \to 0} \frac{1}{\alpha}(1 - I(x)^{-\alpha})$, and thus maximizing the mutual information is equivalent to minimizing the objective function

$$\lim_{\alpha \to 0} \int \frac{\pi(x)}{I^\alpha(x)} \, \mathrm{d}x. \tag{9}$$

The similarities between Eqs. (5), (7), and (9) call for considering the general objective function

$$\int \frac{\pi^a(x)}{I^{p/2}(x)} \, \mathrm{d}x, \tag{10}$$

where $a \in \{1, 2\}$ and $p > 0$. This general objective function includes the three objectives mentioned above. In addition, with $a = 1$, minimizing the objective in Eq. (10) is approximately equivalent to minimizing the $L_p$ reconstruction error, as shown by Morais and Pillow [9]. With $a = 1$ and $p = 1$, it corresponds to minimizing the average discrimination threshold (i.e., the distance needed to be able to distinguish two stimuli that are close). Finally, with $a = 2$ and $p = 1$, Eq. (10) is the expected loss in a discrimination task with constant rewards (see Supplementary Materials).

## 3.2 Optimal encodings under a constraint

The general objective function just presented (Eq. (10)) captures a large gamut of different objectives. In all cases, a larger Fisher information results in improved decisions. We assume, however, that a constraint prevents the observer's brain from choosing an unbounded Fisher information, i.e., that its encoding capacity is limited. Specifically, we call 'efficient' an encoding that solves the problem

$$\min_{I(x)} \int \frac{\pi^a(x)}{I^{p/2}(x)} \, \mathrm{d}x \quad \text{s.t.} \quad \int I^{q/2}(x) \, \mathrm{d}x \leq K^{q/2}, \tag{11}$$

where $a \in \{1, 2\}$, $p > 0$, $q > 0$, and $K > 0$. The constraint on the encoding capacity, in Eq. (11), is the same as in Ref. [9], but we minimize a more general objective function. The solution to this optimum problem is

$$I(x) = K \left( \frac{\pi^{\gamma/2}(x)}{\int \pi^{\gamma/2}(\tilde{x}) \, \mathrm{d}\tilde{x}} \right)^2, \quad \text{where } \gamma = \frac{2a}{p + q}. \tag{12}$$

Thus in all the tasks mentioned above, the optimal Fisher information is proportional to a power of the prior ($I(x) \propto \pi^\gamma(x)$), i.e., efficient coding allocates greater precision to the more frequent stimuli, although the exact allocation depends on the specifics of the objective function [22, 23, 25–27] and of the encoding constraint. Henceforth, we set $q = 1$. The resulting constraint is one that arises naturally in a noisy communication channel (see Supplementary Materials); it is the constraint used in Ref. [6].

## 4 Behavior of the Bayesian mean with different efficient codings

The Equation (12) just presented implies that the relative variation of the Fisher information is proportional to the relative variation of the prior, as

$$\frac{I'(x)}{I(x)} = \gamma \frac{\pi'(x)}{\pi(x)}. \tag{13}$$

In other words, the 'encoding effect', as we have called it above, is predicted to be proportional to the 'prior effect', and the former is larger than the latter if $\gamma > 1$. As mentioned, the relative strength of these two effects determines the direction of the bias. Substituting Eq. (13) in our approximation of the bias (Eq. (3)) results in

$$b(x) = (1 - \gamma) \frac{\pi'(x)}{\pi(x)} \frac{1}{I(x)}, \tag{14}$$

which makes predicting the sign of the bias straightforward: if $\gamma < 1$, the bias is directed toward stimuli of increasing probability, while if $\gamma > 1$, it is directed toward stimuli of decreasing probability (and if $\gamma = 1$, there is no bias).

Before relating these predictions to the corresponding objective functions, we look at the variance of the Bayesian mean and how it relates to its bias, when the encoding is of the form prescribed by Eq. (12). The variance of the Bayesian mean is approximately equal to the inverse of the Fisher information (Eq. (4)), which is proportional to a power of the prior (Eq. (12)), thus the variance of the Bayesian mean is inversely proportional to the same power of the prior, i.e.,

$$v(x) \propto \frac{1}{\pi^\gamma(x)}. \tag{15}$$

An efficient coding of the stimuli thus results in a greater variability of the Bayesian mean for the stimuli that are less frequent. Furthermore, the bias can be expressed as a linear function of the derivative of the inverse of the Fisher information (e.g., substitute in Eq. (14) the relative variation of the prior implied by Eq. (13)), i.e., as a linear function of the derivative of the variance, $v'(x)$. Specifically,

$$b(x) = \frac{\gamma - 1}{\gamma} v'(x). \tag{16}$$

The bias of the Bayesian mean is thus predicted to be proportional to the derivative of its variance. We note that the relation between these two quantities does not depend on the prior, $\pi(x)$, hence it is predicted even for an observer who holds incorrect beliefs about the distribution of the stimuli (perhaps because of incomplete learning).

Our results show that the direction of the bias, and the value and sign of the constant of proportionality in Eq. (16), depend on the objective function that the encoding stage optimizes, and thus on the task that the observer is facing. For an estimation task with squared-error loss ($\gamma = 2/3$), the bias is in the direction of the more probable stimuli. Conversely, for a discrimination task with proportional rewards ($\gamma = 4/3$), the bias is in the direction of the less probable stimuli. Our observer's decoder, the Bayesian mean, is optimal for the two tasks just mentioned, and thus for these two tasks and the corresponding optimal choices of encoding, the entire encoding-decoding process is consistently optimal; yet the resulting directions of the biases in these two tasks are opposite. This underlines the impact of the choice of encoding on the observer's perceptions. As for the other objectives, we find that minimizing the expected discrimination threshold ($\gamma = 1$) results in the absence of any bias ($b(x) = 0$), and minimizing the expected loss in a discrimination task with constant rewards yields the same encoding as maximizing the mutual information ($\gamma = 2$), which results in a bias toward the less probable stimuli. Table 1 summarizes these results.

Table 1: **The direction of the bias of the Bayesian mean depends on the objective that the encoding stage optimizes.** For each of the encoding objectives (*rows*), values of the corresponding constants ($a$, $p$, and $\gamma$, with $q = 1$), power of the prior which the optimal Fisher information is proportional to ($I(x) \propto$), sign of the bias of the Bayesian mean in comparison to that of the derivative of the prior ($b\pi'$), and relation between the bias, $b(x)$, and the derivative of the variance, $v'(x)$. If $b\pi' < 0$, the bias is 'anti-Bayesian', i.e., in the direction of less probable stimuli. '*' indicates the objectives for which the Bayesian mean is optimal.

| Encoding objective | $a$ | $p$ | $\gamma$ | $I(x) \propto$ | $b\pi'$ | Bias vs. variance |
|---|---|---|---|---|---|---|
| Squared error* | 1 | 2 | 2/3 | $\pi^{2/3}(x)$ | $> 0$ | $b(x) = -\frac{1}{2}v'(x)$ |
| Discrimin. threshold | 1 | 1 | 1 | $\pi(x)$ | $= 0$ | $b(x) = 0$ |
| Discrimin. task with proportional rewards* | 2 | 2 | 4/3 | $\pi^{4/3}(x)$ | $< 0$ | $b(x) = \frac{1}{4}v'(x)$ |
| Discrimin. task with constant rewards | 2 | 1 | 2 | $\pi^2(x)$ | $< 0$ | $b(x) = \frac{1}{2}v'(x)$ |
| Mutual Information $MI(x, r)$ | 1 | 0 | 2 | $\pi^2(x)$ | $< 0$ | $b(x) = \frac{1}{2}v'(x)$ |

## 5   Simulations

To assess the quality of the approximations presented above, we run simulations of an encoding-decoding process, and compare the approximated and true values of the bias and variance of the Bayesian-mean decoder, with different efficient encodings, and under different amounts of imprecision in the encoding. We study a case in which a normally-distributed stimulus is encoded through a

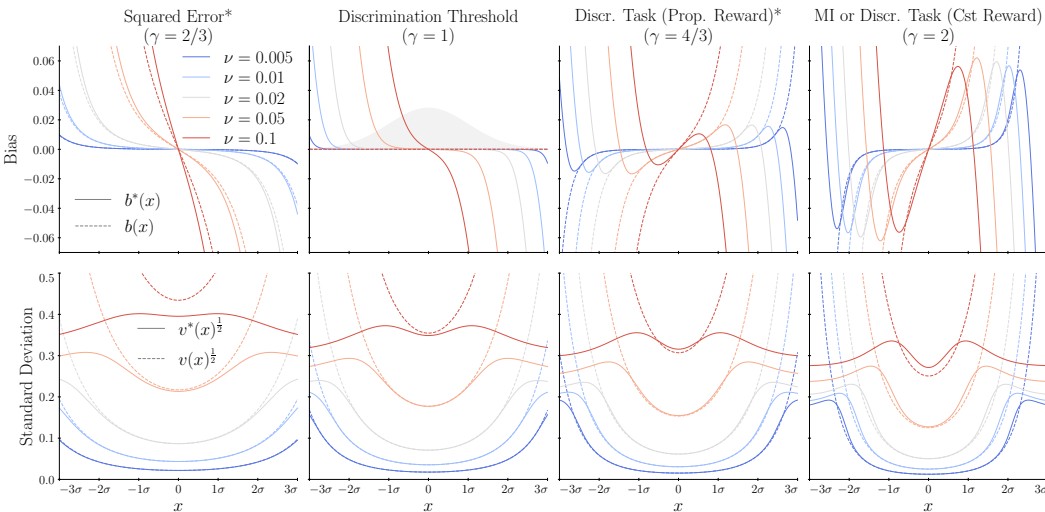

Figure 1: **Bias and variance of the Bayesian mean, and approximations thereof, under different encodings.** *First row:* Bias, $b^*(x)$ (solid lines), and approximation to the bias, $b(x)$ (dashed lines), as a function of the stimulus magnitude, $x$, with different amounts of encoding noise, $\nu$, and with the encoding adapted to four different objectives, each characterized by its constant $\gamma$: 2/3, 1, 4/3, and 2 (columns). The second panel also shows the prior, a standard normal distribution ($m = 0, \sigma = 1$). *Second row:* Standard deviation, $v^*(x)^{\frac{1}{2}}$ (solid lines), and its approximation, $v(x)^{\frac{1}{2}}$ (dashed lines).

normally-distributed representation. Specifically, the prior is a Gaussian distribution with mean $m$ and standard deviation $\sigma$, i.e., $x \sim N(m, \sigma^2)$. Given a stimulus, $x$, the distribution of the encoding internal representation, $r$, is Gaussian, centered on a transformation of the stimulus, $\mu(x)$, and with standard deviation $\nu$ (the 'encoding noise'), i.e., $r|x \sim N(\mu(x), \nu^2)$. The 'transducer' function $x \mapsto \mu(x)$ is an increasing function that maps the stimulus space to the interval $[0, 1]$. The Fisher information of this encoding scheme is $I(x) = (\mu'(x)/\nu)^2$, i.e., the local steepness of the transducer, together with the encoding noise, determine the precision of the encoding. It follows that any encoding rule of this kind necessarily satisfies the constraint in Eq. (11) (with $q = 1$), where the capacity limit is given by $K = 1/\nu^2$. Efficient coding, with a given objective (i.e., with a given $\gamma$), determines the Fisher information, and thus the transducer $\mu(x)$. Under any objective considered here, the optimal encoding and decoding can be written in terms of the centered, scaled stimulus, $(x - m)/\sigma$. Hence without loss of generality we study the case of a standard normal prior distribution, i.e., $m = 0$ and $\sigma = 1$. Finally, note that we have posited a scalar representation, $r$, and not a multi-dimensional vector, because in the case of independent Gaussian signals, a weighted sum of the signals (which will itself be a Gaussian random variable) is a sufficient statistic for the likelihood.

We run simulations in which the encoding noise, $\nu$, spans a range of values: $\nu = 0.005, 0.01, 0.02, 0.05,$ and $0.1$ (these different magnitudes of the encoding noise can be compared to the standard deviation of the prior, $\sigma = 1$). We simulate the encoding process for values of $\gamma$ that correspond to the objective functions mentioned above ($\gamma \in \{2/3, 1, 4/3, 2\}$), including the expected square error ($\gamma = 2/3$, Eq. (5)) and the expected loss in a discrimination task with proportional rewards ($\gamma = 4/3$, Eq. (7)), for which the Bayesian mean is the optimal decoder. In all these cases (characterized by the encoding noise $\nu$ and the constant $\gamma$), we numerically compute the true bias of the Bayesian mean and its variance, which we denote by $b^*(x)$ and $v^*(x)$, and compare them to their analytical approximations, $b(x)$ and $v(x)$.

When the encoding minimizes the expected squared error ($\gamma = 2/3$), and when it minimizes the expected discrimination threshold ($\gamma = 1$), the true bias is in the direction of the mean of the prior, i.e., for positive values of the stimulus, the bias is negative (Fig. 1, first two columns, first row, solid lines). When the encoding minimizes the expected loss in discrimination tasks and when it maximizes the mutual information ($\gamma = 4/3$ or $2$), the stimuli close to the prior mean are perceived with a bias *away* from the mean, and thus toward less probable stimuli. In other words, with these encodings the encoding effect is stronger than the prior effect, and yields 'anti-Bayesian percepts'. Further from

the prior mean, however, the direction of the bias reverses, and points toward the mean (i.e., toward the more probable stimuli; Fig. 1, last two columns, first row, solid lines). Hence when the bias is 'anti-Bayesian' for stimuli close to the mean, it reverses for stimuli farther from the mean. We note that it is not possible to have a 'globally anti-Bayesian' bias: this is not only a feature of the specific examples shown here; it is a general implication of posterior-mean estimation (see Supplementary Materials).

The approximation to the bias is accurate when the stimulus is close to the mean; its quality for a given stimulus depends on the amount of encoding noise, $\nu$. With $\gamma = 2/3$ (squared-error objective), and under small encoding noise ($\nu \leq 0.01$), the error in the approximation is below $10^{-4}$ for any stimulus within two standard deviations ($\pm 2$ s.d.) of the prior mean (see Supplementary Materials). With $\gamma = 1$ (discrimination-threshold objective), the approximation to the bias is zero, while the true bias is different from zero, but small; under small encoding noise ($\nu \leq 0.01$) the error is below $10^{-3}$ for stimuli within $\pm 2$ s.d. of the mean. With $\gamma = 4/3$ and 2 (discrimination-tasks and mutual-information objectives), close to the prior mean the approximation to the bias is correctly directed away from the mean. Further from the mean, the approximation does not reverse (as does the true bias), resulting in large errors. However, within $\pm 2$ s.d. of the prior and under small encoding noise ($\nu \leq 0.01$), the error remains below $10^{-3}$, for $\gamma = 4/3$; and for $\gamma = 2$, the error is below $10^{-2}$ for most stimuli within $\pm 2$ s.d. (see Supplementary Materials). As for the standard deviation of the Bayesian-mean decoder, similarly, the approximation is more accurate for stimuli close to the prior mean, and errors increase with the encoding noise (Fig. 1, second row).

We also look at the true bias of the Bayesian mean, $b^*(x)$, as a function of the derivative of its variance, $\frac{d}{dx} v^*(x)$, in order to examine the accuracy of the relation of proportionality between these quantities that we have derived from their approximations, when the encoding is optimized (Eq. (16)). For stimuli near the prior mean, e.g., within one standard deviation ($\pm 1$ s.d.), and under encoding noise $\nu$ below 0.05, the true bias and derivative of the variance are fairly close to the line prescribed by the relation of proportionality. The slope of this line depends on the constant $\gamma$, i.e., on the task for which the encoding is optimized (see Table 1). For stimuli further from the mean, these two quantities diverge from the linear relation (Fig. 2), but the prediction that they should have the same sign holds over a range of stimuli. For instance, with $\gamma = 2$ and $\nu = 0.02$, the signs are the same for most values of the stimulus within $\pm 2$ s.d. of the prior mean. This weaker prediction implies that we should have an 'anti-Bayesian' bias (when $\gamma > 1$) over the range of stimuli in which the variance increases with the distance from the prior mean.

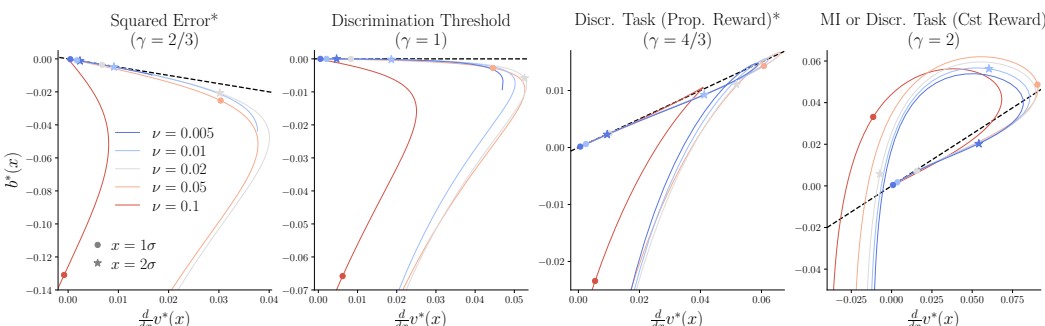

Figure 2: **Linear relation between the bias and the derivative of the variance, close to the prior mean.** Bias, $b^*(x)$, as a function of the derivative of the variance, $\frac{d}{dx} v^*(x)$, for positive values of the stimuli ($x > 0$), with different amounts of encoding noise, $\nu$ (colored lines), and with the encoding adapted to different objectives. Equation (16) predicts a linear relation between the two quantities (dashed line). With $\gamma < 1$, the slope is negative; with $\gamma > 1$, the slope is positive. For $x = 0$, both quantities are zero. The dots and the stars indicate the points at which the stimulus is one and two standards deviations away from the prior mean ($x = 1\sigma$ and $2\sigma$). These curves are symmetric about the origin; values for $x < 0$ are not shown here.

# 6 Discussion

We have presented analytical approximations to the bias and to the variance of the Bayesian-mean decoder, under the assumption that the stimulus is encoded through a series of independent signals, though not necessarily identically distributed. The approximated bias is the sum of two components: one driven by the variations of the prior, which directs the bias towards more probable stimuli, and one driven by the variations of the encoding Fisher information, which directs the bias towards less precisely encoded stimuli (Eq. (3)). When the encoding is efficiently adapted to a given task (such as an estimation task or a discrimination task), these approximations predict a linear relation between the bias and the derivative of the variance (Eq. (16)). The overall direction of the bias depends on the task for which the encoding is optimized (Eq. (14), Table 1). The approximations and the predicted relation are most accurate in the small-noise regime (Fig. 1 and 2).

The approximation to the bias (Eq. (3)) makes explicit the respective roles of the prior and of the encoding in the bias of the Bayesian mean. Traditionally, Bayesian models have been associated to biases directed towards the more probable stimuli [11–14]: in Eq. (3), the term involving the variations of the prior, $\pi'(x)/\pi(x)$, captures this 'prior effect'. The term involving the variations of the Fisher information, $-I'(x)/I(x)$, is directed towards less-precisely encoded stimuli. Indeed, a stimulus that is less precisely-encoded yields imprecise representations, that a more precisely-encoded stimulus would be comparatively unlikely to yield; as a result, the estimated stimulus, decoded from these representations, tends to be "pushed away" from the more precisely-encoded stimulus, and towards the less precisely-encoded stimulus.

This 'encoding effect' is opposite to the 'prior effect', if more probable stimuli are encoded with higher precision (as in efficient-coding models, e.g., Eq. (12)). The ways in which the precision matters depend on the task at hand. Consider, for example, an estimation task with a loss function equal to $|\hat{x} - x|^p$. If $p$ is large, then small errors do not matter greatly; while if $p$ is small, the smallest error in the estimate results in a large loss. Thus with small $p$, the precision with which a stimulus is observed will have a large impact on the task performance; and under a constraint on the encoding capacity, it will be beneficial to allocate more precision to the more frequently occurring stimuli (e.g., $I(x) \propto \pi^2(x)$, in the limit $p \to 0$). By contrast, with large $p$, a small estimation error will not yield a large loss, even for a stimulus that occurs frequently. This results in a different optimal allocation of the precision, with a 'flatter' Fisher information (in the limit $p \to \infty$, $I(x)$ is constant). In other words, the variations of the optimal Fisher information depend on the extent to which small errors are costly. This also applies to other tasks: in a discrimination task with constant rewards, mis-ordering two stimuli that are extremely close results in the maximum possible loss; while in a discrimination task with proportional rewards, it would only result in a loss proportionally as small as the difference between the two stimuli. Consequently, the former task results in an optimal Fisher information that varies more strongly with the prior than that in the latter task ($I(x) \propto \pi^2(x)$ vs. $I(x) \propto \pi^{4/3}(x)$; see Table 1). As for the objective of maximizing the mutual information, it is equivalent to the limiting case $p \to 0$ of the estimation task, i.e., to a task in which all mis-estimations of the stimulus are equally costly, no matter how small the distance between the stimulus and the estimate.

If the 'encoding effect' is stronger than the 'prior effect', the resulting bias is 'anti-Bayesian', i.e., directed towards the less probable stimuli. Such biases have been reported in the literature [6, 15–17]. Although Wei and Stocker [6] show that efficient coding can indeed account for them, 'anti-Bayesian' biases seem at odds with the traditional 'Bayesian' biases mentioned above. An interesting example is provided by Brayanov and Smith [17], who find that when subjects compare the weights of two objects (a *discrimination* task), their perceptions are 'anti-Bayesian'. But in a task involving the *estimation* of weights, the behavior of subjects is consistent with a 'Bayesian' bias, towards prior expectations. Our theoretical results readily explain these seemingly contradictory findings: different tasks imply different efficient codings, which imply different strengths of the 'encoding effect', and thus different biases. Specifically, as shown in Table 1, an estimation task with squared-error loss results in a bias towards more probable stimuli, while the two discrimination tasks we consider yield biases towards less probable stimuli. Our predictions are thus consistent with the empirical results of Brayanov and Smith.

We obtain, moreover, a relation of proportionality between the bias and the derivative of the variance of the Bayesian mean, when the encoding is efficient (Eq. (16)). This prediction is consistent with that of Wei and Stocker [1], who also predict a relation of proportionality, although only when the objective is to maximize the mutual information, which is not the objective in many tasks (moreover,

actual encodings do not seem to be always consistent with this objective, even in early sensory areas [27]). Wei and Stocker call the relation a "law of human perception", and find strong empirical support for it; but the proportionality constant is left unspecified. In our derivation, the encoding optimizes a general objective function (Eq. (10)), which includes the mutual information as a special case, but also includes objectives related to common tasks, such as estimation and discrimination tasks. We predict in addition how the value and sign of the constant of proportionality should depend on the objective that the encoding optimizes. Morais and Pillow [9] present related results: they also consider a generalized objective function, although it does not include discrimination tasks. They derive a similar relation of proportionality in the case of the maximum a posteriori (MAP) decoder, and conjecture a relation in the case of the Bayesian mean. In these studies, however, often both the encoding and decoding stages are optimized to some objectives, but different ones, e.g., the encoding maximizes the mutual information while the decoding minimizes the squared error. Although the mutual information might be a reasonably good 'general-purpose' objective, the empirical results of Brayanov and Smith suggest that the brain is able to optimize different objectives. For estimation tasks with squared-error loss, and for discrimination tasks with proportional rewards, the encoding-decoding scheme we investigate is consistently optimal, i.e., both the encoding and the encoding are optimal for these tasks.

Our derivations rely on the following assumptions: the encoding consists in an accumulation of independent signals; it is optimized for a given task, under the constraint in Eq. (11); the decoder is the Bayesian mean; and the noise is small. Note that the assumptions of optimal encoding and decoding implicitly entails that the decision-maker knows the prior. Violations of these assumptions would distort the results in ways that would be interesting to explore, but that are outside the scope of this paper. The relevance of the Bayesian approach, for instance, has been a matter of debate (see Refs. [30–33]). As for the independence assumption, while noise in the activities of neurons in the brain is typically correlated [28, 29], our derivations rely on a central limit theorem, and we surmise that they should withstand some amount of correlation in the encoding.

Overall, our results suggest a parsimonious theory of perception, which mainly rests on assumptions of optimality and on a constraint on the encoding capacity. Under this theory, just three elements shape the statistics of percepts: the prior, the specifics of the task, and the encoding constraint.

We do not see any obvious, potentially negative societal impacts of our research. Indirectly, a better understanding of human perception could be used for deceptive purposes; we deem such an application to be very speculative at this stage.

## Acknowledgments and Disclosure of Funding

This work was supported by a grant from the National Science Foundation (SES DRMS 1949418). A.P.C. was supported by a Fellowship from the Italian Academy for Advanced Studies in America at Columbia University.

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
