# Bias and variance of the Bayesian-mean decoder
## SUPPLEMENTARY MATERIAL

**Arthur Prat-Carrabin**
Department of Economics
Columbia University
New York, USA
arthur.p@columbia.edu

**Michael Woodford**
Department of Economics
Columbia University
New York, USA
mw2230@columbia.edu

## A  Asymptotic approximations

To derive the bias of the Bayesian mean, we first derive the bias of the maximum-likelihood estimator (MLE), after which we express the Bayesian mean as a function of the MLE. We assume that the prior, $\pi(x)$, and the likelihoods of the internal signals, $x \mapsto f_i(r_i|x)$, are smooth functions, so that all the derivatives taken below are well defined.

### A.1  Bias of the maximum-likelihood estimator

Here we present the main steps of a derivation of the bias of the MLE. Our derivation is directly inspired by Cox and Snell (1968) (Ref. [21]). Let $x$ be the stimulus. As in the main text, $r = (r_1, \ldots, r_n)$ is a vector of $n$ samples drawn independently from $n$ distributions, $f_i(r_i|x)$. We use the following notations:

$$L_i(x) = \ln f_i(r_i|x), \tag{17}$$

$$\text{and } L(x) = \sum_{i=1}^{n} L_i(x). \tag{18}$$

We start with well-known results. First, we necessarily have $\mathbb{E}L_i'(x) = 0$ at any point $x$. Second, the Fisher information of the variable $r_i$ is:

$$I_i(x) = \text{Var } L_i'(x) = \mathbb{E}[L_i'(x)^2] = -\mathbb{E}L_i''(x), \tag{19}$$

and the Fisher information of the vector $r$ is

$$I(x) = \text{Var } L'(x) = \sum_{i=1}^{n} I_i(x) = \mathbb{E}[L'(x)^2] = -\mathbb{E}L''(x). \tag{20}$$

We assume that the sequences of random variables $(L_1'(x), \ldots, L_n'(x))$ and $(L_1''(x), \ldots, L_n''(x))$ satisfy the conditions of the Lyapunov central limit theorem[1]. The central limit theorem provides convergence results for the sums $L'(x) = \sum L_i'(x)$ and $L''(x) = \sum L_i''(x)$, as $n$ goes to infinity:

$$\frac{L'(x)}{\sqrt{I(x)}} \xrightarrow{d} N(0,1), \tag{21}$$

---

[1] Lyapunov central limit theorem: let $(X_1, \ldots, X_n)$ be a sequence of independent random variables with finite expectation and variance, and let $X = \sum_i X_i$ and $s_n^2 = \sum_i \text{Var } X_i$. If there exists $\delta > 0$ such that $\lim_{n\to\infty} \frac{1}{s_n^{2+\delta}} \sum_i \mathbb{E}[|X_i - \mathbb{E}X_i|^{2+\delta}] = 0$, then $(X - \mathbb{E}X)/s_n$ converges in distribution to the standard normal distribution, $N(0,1)$.

35th Conference on Neural Information Processing Systems (NeurIPS 2021).

and

$$\frac{L''(x) + I(x)}{\text{Var } L''(x)} \xrightarrow{d} N(0, 1). \tag{22}$$

Thus

$$L'(x) = O(\sqrt{n}) \tag{23}$$

and

$$L''(x) = -I(x) + O(\sqrt{n}). \tag{24}$$

Note that $I(x)$ and $L(x)$ and its derivatives are of order $n$.

Let $x^*$ be the maximum-likelihood estimator. We have $L'(x^*) = 0$, and thus approximately

$$L'(x) + (x^* - x)L''(x) = 0. \tag{25}$$

This results in

$$x^* - x = \frac{L'(x)}{-L''(x)} = \frac{L'(x)}{I(x)} + O\left(\frac{1}{n}\right). \tag{26}$$

Thus

$$\mathbb{E}(x^* - x) = O\left(\frac{1}{n}\right), \tag{27}$$

and

$$\mathbb{E}(x^* - x)^2 = \frac{1}{I(x)} + O\left(\frac{1}{n^{3/2}}\right). \tag{28}$$

We now expand $L'(x^*)$ to a higher order:

$$L'(x) + (x^* - x)L''(x) + \frac{1}{2}(x^* - x)^2 L'''(x) = 0, \tag{29}$$

where $L'''$ is the third derivative of $L$. Taking the expected value, we obtain

$$\mathbb{E}(x^* - x)\mathbb{E}L''(x) + \text{Cov}\left(x^* - x, L''(x)\right) + \frac{1}{2}\mathbb{E}(x^* - x)^2 \mathbb{E}L'''(x) + \frac{1}{2}\text{Cov}\left((x^* - x)^2, L'''(x)\right) = 0. \tag{30}$$

We examine the orders of magnitude, in terms of powers of $n$, of the elements in this equation. We find (dropping temporarily the dependence on $x$ of $L$ and $I$)

$$\text{Cov}\left(x^* - x, L''\right) = \frac{1}{I}\mathbb{E}L'L'' + O\left(\frac{1}{\sqrt{n}}\right), \tag{31}$$

$$\text{Cov}\left((x^* - x)^2, L'''\right) = O\left(\frac{1}{n}\right), \tag{32}$$

$$\text{and } \mathbb{E}(x^* - x)^2 \mathbb{E}L''' = \frac{1}{I}\mathbb{E}L''' + O\left(\frac{1}{\sqrt{n}}\right). \tag{33}$$

Substituting these relations in Eq. (30) we have

$$-I\mathbb{E}(x^* - x) + \frac{1}{I}\mathbb{E}L'L'' + \frac{1}{2I}\mathbb{E}L''' + O\left(\frac{1}{\sqrt{n}}\right) = 0, \tag{34}$$

and thus

$$\mathbb{E}(x^* - x) = \frac{1}{I^2}\left(\mathbb{E}L'L'' + \frac{1}{2}\mathbb{E}L'''\right) + O\left(\frac{1}{n^{3/2}}\right). \tag{35}$$

We note that $\mathbb{E}L''' = \sum_i \mathbb{E}L_i'''$, and, as the samples are independently drawn, $\mathbb{E}L'L'' = \sum_i \mathbb{E}L_i'L_i''$. Moreover, just as with the relation $\mathbb{E}[L_i'(x)^2] = -\mathbb{E}L_i''(x)$, there are relations between the means of higher powers and higher derivatives of $L_i$. In particular, one can show that

$$\mathbb{E}[L_i'(x)L_i''(x)] = \frac{1}{2}[I_i'(x) - J_i(x)], \tag{36}$$

where $I_i'(x)$ is the derivative of $I_i(x)$ and $J_i(x) = \mathbb{E}[L_i'(x)^3]$. Another relation is

$$\mathbb{E}[L_i'''(x)] = \frac{1}{2}J_i(x) - \frac{3}{2}I_i'(x). \tag{37}$$

Substituting these relations in Eq. (35) results in the following expression for the bias of the maximum-likelihood estimator:

$$\mathbb{E}(x^* - x) = -\frac{I'(x) + J(x)}{4I^2(x)} + O\left(\frac{1}{n^{3/2}}\right), \tag{38}$$

where $J(x) = \sum_i J_i(x) = \mathbb{E}[L'(x)^3]$.

## A.2 Bias of the Bayesian mean

As shown in Eq. (2), the Bayesian mean is the ratio of two integrals of the type $\int g(\tilde{x})e^{L(\tilde{x})}\,\mathrm{d}\tilde{x}$, which is amenable to a Laplace approximation. Let $\varepsilon = \tilde{x} - x^*$. Taylor expansions of $L(\tilde{x})$ and $g(\tilde{x})$ at $x^*$ yield the approximations

$$L(\tilde{x}) = L(x^*) + \frac{1}{2}L''(x^*)\varepsilon^2 + \frac{1}{6}L'''(x^*)\varepsilon^3 + \frac{1}{24}L^{(4)}(x^*)\varepsilon^4 + O(n\varepsilon^5), \tag{39}$$

and thus

$$e^{L(\tilde{x})} = e^{L(x^*)}e^{\frac{1}{2}L''(x^*)\varepsilon^2}\left(1 + \frac{1}{6}L'''(x^*)\varepsilon^3 + \frac{1}{24}L^{(4)}(x^*)\varepsilon^4 + O(n\varepsilon^5)\right), \tag{40}$$

and

$$g(\tilde{x}) = g(x^*) + g'(x^*)\varepsilon + \frac{1}{2}g''(x^*)\varepsilon^2 + \frac{1}{6}g'''(x^*)\varepsilon^3 + \frac{1}{24}g^{(4)}(x^*)\varepsilon^4 + O(\varepsilon^5), \tag{41}$$

where $L^{(4)}$ if the fourth derivative of $L$, and $g'$, $g''$, $g'''$ and $g^{(4)}$ are the first to fourth derivatives of $g$. The product of the right-hand sides of the last two equations is a polynomial of $\varepsilon$ multiplied by a Gaussian function of $\varepsilon$. Taking the integral of this product, we find

$$\int g(\tilde{x})e^{L(\tilde{x})}\,\mathrm{d}\tilde{x} = \left[g(x^*) + \frac{g''(x^*)}{-2L''(x^*)} + \frac{\frac{1}{4}g(x^*)L^{(4)}(x^*) + g'(x^*)L'''(x^*)}{2(L''(x^*))^2} + O(\frac{1}{n^2})\right]\frac{e^{L(x^*)}\sqrt{2\pi}}{\sqrt{|L''(x^*)|}}. \tag{42}$$

We use this approximation in the expression of the Bayesian mean (Eq. (2)), with $g(x) = x\pi(x)$ for the numerator, and $g(x) = \pi(x)$ for the denominator. We obtain an expression of the Bayesian mean as a function of the MLE, as

$$\hat{x} = x^* + \frac{1}{-L''(x^*)}\frac{\pi'(x^*)}{\pi(x^*)} + \frac{1}{2}\frac{L'''(x^*)}{(L''(x^*))^2} + O(1/n^2). \tag{43}$$

This equation is consistent with the result reported by Ref. [34]. We can further approximate the right-hand side by taking the functions involved at the point $x$ instead of at the MLE $x^*$. We have

$$L''(x^*) = L''(x) + O(\sqrt{n}), \tag{44}$$

$$\frac{1}{-L''(x^*)} = \frac{1}{L''(x)} + O(1/n^{3/2}), \tag{45}$$

$$\frac{\pi'(x^*)}{\pi(x^*)} = \frac{\pi'(x)}{\pi(x)} + O(1/\sqrt{n}), \tag{46}$$

$$\text{and } \frac{L'''(x^*)}{(L''(x^*))^2} = \frac{L'''(x)}{(L''(x))^2} + O(1/n^{3/2}), \tag{47}$$

thus

$$\hat{x} = x^* + \frac{1}{-L''(x)}\frac{\pi'(x)}{\pi(x)} + \frac{1}{2}\frac{L'''(x)}{(L''(x))^2} + O(1/n^{3/2}). \tag{48}$$

In addition, using Eq. (24),

$$\frac{1}{L''(x)} = -\frac{1}{I(x)} + O(1/n^{3/2}), \tag{49}$$

$$\text{and } \frac{L'''(x)}{(L''(x))^2} = \frac{\mathbb{E}L'''(x)}{I^2(x)} + O(1/n^{3/2}), \tag{50}$$

thus

$$\hat{x} = x^* + \frac{1}{I(x)}\frac{\pi'(x)}{\pi(x)} + \frac{1}{2}\frac{\mathbb{E}L'''(x)}{I^2(x)} + O(1/n^{3/2}). \tag{51}$$

We have thus written the Bayesian mean, $\hat{x}$, as the MLE, $x^*$, corrected by a function of the true stimulus, $x$. Substituting $\mathbb{E}L'''(x)$ (see Eq. (37)), we obtain

$$\hat{x} = x^* + \frac{1}{I(x)}\frac{\pi'(x)}{\pi(x)} + \frac{1}{4}\frac{J(x)}{I^2(x)} - \frac{3}{4}\frac{I'(x)}{I^2(x)} + O(1/n^{3/2}). \tag{52}$$

Using the expression of the bias of the MLE, derived above (Eq. (38)), we obtain, in expectation,

$$\mathbb{E}\hat{x} = x + \frac{1}{I(x)}\frac{\pi'(x)}{\pi(x)} - \frac{I'(x)}{I^2(x)} + O(1/n^{3/2}). \tag{53}$$

The second and third term constitute our approximation to the bias of the Bayesian mean, $b(x)$ (Eq. (3)). Equation (53) is consistent with the result (3.10) of Ref. [18].

### A.3 Variance and expected squared error of the Bayesian mean

Using Eqs. (52) and (53), we derive an approximation to the squared deviation of $\hat{x}$ from its mean, as

$$(\hat{x} - \mathbb{E}\hat{x})^2 = (x^* - x)^2 + O(1/n^{3/2}). \tag{54}$$

(The orders of additional terms are powers of $1/n$ greater than or equal to $3/2$.) From Eq. (28) it follows that the variance of the Bayesian mean is

$$\mathbb{E}(\hat{x} - \mathbb{E}\hat{x})^2 = \frac{1}{I(x)} + O(1/n^{3/2}). \tag{55}$$

Finally, we consider the expected squared error of the Bayesian mean,

$$\mathbb{E}(\hat{x} - x)^2 = \mathbb{E}(\hat{x} - x^*)^2 + 2\mathbb{E}(\hat{x} - x^*)(x^* - x) + \mathbb{E}(x^* - x)^2. \tag{56}$$

The first two terms are of order $1/n^2$, and the third is given by Eq. (28). Thus

$$\mathbb{E}(\hat{x} - x)^2 = \frac{1}{I(x)} + O(1/n^{3/2}). \tag{57}$$

## B   The bias of the Bayesian mean cannot be 'globally anti-Bayesian'

The posterior-mean decoder assigns to any internal representation $r$ the decoded value $\hat{x}(r) \equiv \mathbb{E}[x|r]$, where the expectation is over the joint distribution for $x$ and $r$ implied by the prior $\pi$ and the encoding rule. This decoding rule, together with the prior and the encoding rule, then defines a joint distribution for $x, r$ and $\hat{x}$. The law of iterated expectations implies that $\mathbb{E}[\mathbb{E}[x|r]\,|\hat{x}(r) = \hat{x}] = \mathbb{E}[x|\hat{x}]$, where the outer expectation on the left integrates over the marginal joint distribution for $r$ and $\hat{x}$, and the expectation on the right integrates over the marginal joint distribution for $x$ and $\hat{x}$. Hence posterior-mean decoding requires that

$$\mathbb{E}[x\,|\hat{x}] = \hat{x} \tag{58}$$

for all $\hat{x}$, regardless of the nature of the prior and the encoding rule.

We will say that the bias function $b^*(x)$ is *globally anti-Bayesian* if there exists some measure $\bar{x}$ of the central tendency under the prior (this might be the prior mode, but it could also be the prior median, the prior mean, the geometric mean, or any other point in the interior of the support) such that $b^*(x) < 0$ for all $x < \bar{x}$ and $b^*(x) > 0$ for all $x > \bar{x}$ (so that the estimate is repelled from the value $\bar{x}$). Note that in the case of a unimodal prior (one with $\pi'(x) > 0$ for all $x < \bar{x}$ and $\pi'(x) < 0$ for all $x > \bar{x}$, where $\bar{x}$ is the mode), and the encoding rule (12), our asymptotic approximations (15)–(16) imply that the bias function should be globally anti-Bayesian whenever $\gamma > 1$.

But this property is inconsistent with Eq. (58). Suppose that the bias is globally anti-Bayesian. It follows that $\mathbb{E}[\hat{x} - \bar{x}\,|x]$ must be greater than $x - \bar{x}$ whenever the latter quantity is positive, and smaller than it whenever the latter quantity is negative; hence one must have

$$\mathbb{E}[(\hat{x} - \bar{x})^2\,|x] > (x - \bar{x})^2 \tag{59}$$

for any $x \neq \bar{x}$. Taking the expectation of both sides of this inequality under the prior for $x$, and again using the law of iterated expectations, we see that we must have

$$\mathbb{E}[(\hat{x} - \bar{x})^2] = \mathbb{E}[\mathbb{E}[(\hat{x} - \bar{x})^2\,|x]] > \mathbb{E}[(x - \bar{x})^2] \tag{60}$$

in the case of any prior that does not place the entire probability mass at $x = \bar{x}$.

At the same time, posterior-mean decoding requires that we must have

$$\mathbb{E}[(x - \bar{x})^2\,|\hat{x}] = (\mathbb{E}[x - \bar{x}\,|\hat{x}])^2 + \mathrm{Var}[x - \bar{x}\,|\hat{x}] \tag{61}$$

$$= (\hat{x} - \bar{x})^2 + \mathrm{Var}[x|\hat{x}] \tag{62}$$

as a consequence of Eq. (58). Taking the expectation of both sides (integrating over the marginal distribution for $\hat{x}$), and again using the law of iterated expectations, we see that we must have

$$\mathbb{E}[(x - \bar{x})^2] = \mathbb{E}[\mathbb{E}[(x - \bar{x})^2\,|\hat{x}]] = \mathbb{E}[(\hat{x} - \bar{x})^2] + \mathbb{E}[\mathrm{Var}[x|\hat{x}]]. \tag{63}$$

But this implies that we must have

$$\mathbb{E}[(x - \bar{x})^2] \geq \mathbb{E}[(\hat{x} - \bar{x})^2], \tag{64}$$

contradicting Eq. (60).

Thus the bias cannot have the assumed (anti-Bayesian) sign for all $x$ in the support of the prior, assuming that $x \neq \bar{x}$ with positive probability. The numerical solutions shown in Figure 1 for the cases with $\gamma > 1$ do not violate this prediction, even when $\nu$ is small. When $\nu = 0.005$, we see in each of the last two columns that the bias is anti-Bayesian (and closely tracks the prediction of the asymptotic approximation) for all values of $x$ within two standard deviations of the prior mean (which is also the prior mode); but whereas the asymptotic approximation predicts that the bias should continue to be anti-Bayesian even for more extreme values of $x$, our numerical solutions indicate a sharp change in the sign of the bias for more extreme values of $x$. When $\nu$ is larger, the anti-Bayesian bias is even stronger for values of $x$ near the prior mean, but the range of values of $x$ for which the bias is anti-Bayesian is smaller (because the range of values for which the asymptotic approximation is reliable shrinks).

The validity of the asymptotic approximation for small enough values of $\nu$ means that for any value of $x$, the value of $b^*(x)$ approaches the value $b(x)$ given by the asymptotic approximation for all small enough values of $\nu$. In the case of a unimodal prior (as assumed in Figure 1), this means that $b^*(x)$ has the anti-Bayesian sign for all small enough values of $\nu$. However, this does not mean that the bias will be globally anti-Bayesian for any value of $\nu$, since the convergence of $b^*(x)$ to the globally anti-Bayesian function $b(x)$ is not uniform in $x$. Our numerical results indicate that $b^*(x)$ converges relatively rapidly to $b(x)$ for values of $x$ near the prior mean, but more and more slowly for progressively more extreme values of $x$.

## C   Encoding objective functions

To approximate the expected loss in a discrimination task with proportional rewards (Eq. (6)), we note that asymptotically, the MLE is normally distributed (see Eqs. (21) and (26)), and that it provides an approximation to the Bayesian mean (see Eq. (52)). We thus approximate the distribution of the Bayesian-mean estimator $\hat{x}_i$ of the stimulus $x_i$ by a Gaussian distribution with mean $x_i$ and variance $1/I(x_i)$. The difference between two estimates, $\hat{x}_1 - \hat{x}_0$, is then normally distributed around $x_1 - x_0$, with variance $1/I(x_0) + 1/I(x_1)$. With

$$z = \frac{x_1 - x_0}{\sqrt{\frac{1}{I(x_0)} + \frac{1}{I(x_1)}}}, \tag{65}$$

and denoting by $\Phi$ the standard normal CDF, the probability of erroneously ordering the two stimuli is $\Phi(z)$ if $z < 0$, and $\Phi(-z)$ if $z > 0$, i.e., $P(\text{error}|x_0, x_1) = \Phi(-|z|)$. Fixing $x_0$ (in Eq. (6)), the expected loss averaged over $x_1$ is approximately

$$\int \Phi(-|z|)|z| \Big( \frac{1}{I(x_0)} + \frac{1}{I(x_1)} \Big) \pi(x_1) \, dz \tag{66}$$

$$\simeq \frac{\pi(x_0)}{I(x_0)} \int 2\Phi(-|z|)|z| \, dz \tag{67}$$

$$\propto \frac{\pi(x_0)}{I(x_0)}. \tag{68}$$

The expected loss in a discrimination task with proportional rewards (Eq. (6)) is thus proportional to this quantity averaged over the distribution of $x_0$, i.e., it to the quantity in Eq. (7).

As for the expected loss with constant rewards, it is proportional to

$$\iint P(\text{error}|x_0, x_1)\pi(x_0)\pi(x_1) \, dx_0 \, dx_1. \tag{69}$$

With the same approximations as those presented just above, it is straightforward to show that this quantity is proportional to the approximate expected loss

$$\int \frac{\pi^2(x)}{\sqrt{I(x)}} \, dx, \tag{70}$$

i.e., Eq. (10) with $a = 2$ and $p = 1$.

Finally, the discrimination threshold is the difference $\delta_{x_0}$ between two close stimuli $x_0$ and $x_1 = x_0 + \delta_{x_0}$ such that the probability of correctly distinguishing the two is above some given success rate. We approximate the probability of error, as

$$P(\text{error}|x_0, x_1 = x_0 + \delta_{x_0}) = \Phi\left(\frac{-|\delta_{x_0}|}{\sqrt{\frac{1}{I(x_0)} + \frac{1}{I(x_0 + \delta_{x_0})}}}\right) \tag{71}$$

$$\simeq \Phi\left(-|\delta_{x_0}|\sqrt{\frac{I(x_0)}{2}}\right) \tag{72}$$

$$\simeq \frac{1}{2} - |\delta_{x_0}|\sqrt{\frac{I(x_0)}{2}} \Phi'(0). \tag{73}$$

The discrimination threshold is thus proportional to $1/\sqrt{I(x_0)}$, and its average over the distribution of stimuli $x_0$ is proportional to:

$$\int \frac{\pi(x_0)}{\sqrt{I(x_0)}} \, \mathrm{d}x_0, \tag{74}$$

i.e., Eq. (10) with $a = 1$ and $p = 1$.

## D   Encoding constraint

The specific constraint on the Fisher information that we consider, $q = 1$ in Eq. (11), has been used in the literature, in particular in Ref. [6] for a similar optimization problem. This constraint on the possible variation in the Fisher information over the sensory space arises in any model where one assumes that information must be transmitted through a communication channel that produces a noisy output signal $r$ when an input signal $m$ is supplied, where $m$ (like the physical feature to be encoded) is unidimensional, though $r$ need not be.

Suppose that the conditional probabilities $p(r|m)$ are given by the biophysics of the channel, but that both the way that physical states $x$ are encoded as inputs to the channel (i.e., the encoding function $m = m(x)$) and the way that an estimate of the physical state is produced using the output signal (i.e., the decoding function $\hat{x} = \hat{x}(r)$) are to be optimized for a given environment (prior distribution over states $x$) and decision problem. Then in the case of any differentiable, monotonically increasing encoding function $m(x)$, the Fisher information at $x$ is equal to

$$I(x) = \langle(\frac{\partial \log p(r|m(x))}{\partial x})^2\rangle = \langle(\frac{\partial \log p(r|m(x))}{\partial m} \cdot m'(x))^2\rangle = m'(x)^2 \cdot I^*(m(x)),$$

where $I^*(m)$ is the Fisher information of the channel at input signal $m$ (which is independent of the encoding rule). By reparameterizing the input space (which is innocuous if we allow arbitrary monotonic encoding rules), we can choose a measure of $m$ such that $I^*(m) = k^2 > 0$ over the entire input range of the channel. It then follows that $(I(x))^{1/2} = km'(x)$, so that for any monotonic encoding rule, $\int (I(x))^{1/2}dx = k(\bar{m} - \underline{m})$, where $\underline{m}$ and $\bar{m}$ are the lower and upper bounds of the range of the encoding function. If the properties of the channel require $m$ to remain within a bounded range, then this range (called the "Thurstone invariant" in Ref. [35]) determines a finite upper bound for $\int (I(x))^{1/2}dx$.

As a concrete example of such a coding problem, we assume in our calculations that the internal representation $r$ is a real number, drawn from a Gaussian distribution $N(m(x), \nu^2)$, where $\nu$ is independent of the input $m(x)$. This is an example of a communication channel for which the Fisher information is uniform over the channel input space, $I^*(m) = \nu^{-2}$ for all $m$. As a more biophysically realistic example, Ganguli and Simoncelli [36] propose a model with a heterogeneous population of neurons whose tuning curves "tile" the line corresponding to different values of $m$, a nonlinear transformation of the stimulus space (where the nonlinear transformation is to be optimized). In this example, $r$ is high-dimensional (it corresponds to a vector of spike counts by the different neurons in the population), but the probabilities $p(r|m)$ are taken as given, and moreover satisfy a translational invariance that imply (as in our simpler model) that $I^*(m)$ is uniform over the input space $m$. The finiteness of the range of possible values of $m$ for which this uniform Fisher information can be maintained then follows from the finiteness of the population of neurons (Ganguli and Simoncelli link the Fisher information of their channel to the local density of the population of neurons in

each range of values of $m$). Ref. [7] derives the same form of resource constraint in a model of a neuron whose firing rate follows a sigmoidal tuning curve; here the input signal $m$ corresponds to a monotonic transformation of the neuron's firing rate (so that the encoding function $m(x)$ is effectively the neuron's tuning curve), while the output signal $r$ is the number of spikes produced. In this case, the bounded range for $m$ reflects a bounded range of possible firing rates for the neuron.

In addition, Wei and Stocker [37] argue that the quantity bounded in this constraint is proportional to the number of different stimuli that the encoding is able to discriminate, thus providing a measure of the encoding capacity of a neural system. They note, moreover, that this formulation of the resource constraint has the attractive feature of being invariant under reparameterization of $x$.

## E Errors in the approximations

Figure 3 shows in logarithmic scale the absolute error of the approximations to the bias and to the variance of the Bayesian mean.

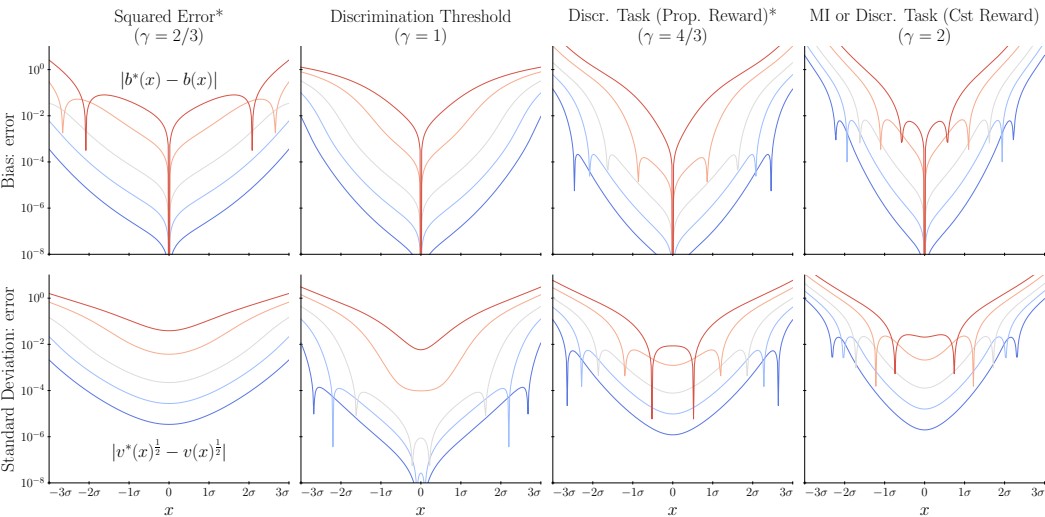

Figure 3: **Errors in the approximations of the bias and the variance of the Bayesian mean, under different encodings.** *First row:* Absolute difference between the bias and its approximation, $|b^*(x) - b(x)|$, as a function of the stimulus magnitude, $x$, with different amounts of encoding noise, $\nu$, and with the encoding adapted to four different objectives, each characterized by its constant $\gamma$: 2/3, 1, 4/3, and 2 (first to last column). *Second row:* Absolute difference between the standard deviation and its approximation, $|v^*(x)^{\frac{1}{2}} - v(x)^{\frac{1}{2}}|$.