# OpenReview forum: "Bias and variance of the Bayesian-mean decoder"
_NeurIPS.cc/2021/Conference — NeurIPS 2021 Spotlight_

### Official Review · Reviewer_v6Bb · 2021-07-15

**Rating:** 6
**Confidence:** 2

**Summary:**

The paper investigates the bias (an the variance) of the Bayesian-mean decoder. The authors found that the bias is driven by two components: The variations of the priors and variations of the Fisher information, and the exact relation is driven by the encoding objective. Simulations are used to back the results.

Disclaimer: I am an experimental visual psychophysicist with experience in Deep Learning. Thus the paper falls a bit outside my area of expertise (as indicated to the AC). I am focusing on the general aspects of this paper.

**Limitations And Societal Impact:**

The authors describe the limitations of the work implicitly. Although, they are never explicitly stated.

**Main Review:**

Quality:

I think this paper could be of great interest to the experimental/psychophysical community in general. However, I am missing one crucial point here; that is a comparison to experimental data. A comparison to the experimental data would make the paper much stronger. Right now, the mathematical details make an accurate impression, but without empirical support, it is difficult to judge the evidence for this theory. For this purpose, the authors could easily use existing datasets.

From my point of view, using empirical data would make the paper much more interesting; for example, if one determines the mean and variance for some stimuli experimentally, one can draw conclusions about the corresponding coding strategy. Vice versa, one could try to enforce different encoding strategies and verify the predicted relation of mean and variance.

Clarity:
Overall, the quality of the paper is very high. The previous work is presented clearly. Also the mathematical parts are well structured.

**Time Spent Reviewing:**

5

---

> ### Author Response · Authors · 2021-08-09
> **Response to Reviewer v6Bb**
>
> We thank Reviewer v6Bb for her/his positive comments on our work.
>
> We agree that the analysis of empirical data would be an interesting addition, to the extent that the page limit permits it. We are, in fact, working on experiments very much in the spirit of what the Reviewer suggests. As for existing datasets, we have found that, when they are publicly available, they are often not quite appropriate for a quantitative analysis aimed at examining the relevance of our theory. Estimation tasks, for instance, would allow directly determining the mean and the variance, as suggested by the Reviewer. They are not, however, as common as discrimination tasks, and often the objective and the prior are not precisely stated (and the actual distribution of stimuli used in the experiment does not correspond to the natural distribution of similar stimuli), so that it is not clear how the encoding and the decoding should be optimized. Moreover, the theory predicts that the encoding and decoding should adapt to the objective of the task and to the prior, but psychophysics experiments typically do not manipulate these parameters. The experiments of Brayanov and Smith (Ref. [17]), on which we comment in the paper, are an exception: the authors find anti-Bayesian biases in a discrimination task, and “Bayesian” biases in an implicit estimation task, consistently with our theory (we refer to our response to Reviewer 5yDC for more details on these tasks).

---

### Official Review · Reviewer_s1gn · 2021-07-17

**Rating:** 6
**Confidence:** 3

**Summary:**

This is theoretical contribution investigating Wei and Stocker's so-called “law of human perception” (ref 1). The authors generalize the "law" to more include general loss functions inspired by Morais and Pillow (ref 9).

**Ethical Concerns:**

No ethical concerns

**Limitations And Societal Impact:**

See above - no adversarial impact on society detected

**Main Review:**

I rather enjoyed reading this theoretical paper, for its clarity and rigor of the exposition.
Originality, however, could be higher. The main results are derived combining two existing and well known papers (ref 1 and 9)

The quality of the paper is good, technical derivations are detailed enough to follow and the paper does a good job of situating the contribution among  a fair a relevant set of references.  The main objection to the Bayesian approach are not included e.g. the critique offered by Bowers, J.S. and Davis, C.J., 2012. Bayesian just-so stories in psychology and neuroscience. Psychological bulletin, 138(3), p.389.    See also the well-tempered rebuttal of this critique in Griffiths, T.L., Chater, N., Norris, D. and Pouget, A., 2012. How the Bayesians got their beliefs (and what those beliefs actually are): comment on Bowers and Davis (2012).

Clarity is good. The paper is well written.

Significance is mainly in reviewing and putting the works of ref 1 and ref 9 in a joint exposure.

**Time Spent Reviewing:**

3

---

> ### Author Response · Authors · 2021-08-09
> **Response to Reviewer s1gn**
>
> We thank Reviewer s1gn for her/his positive comments on our work, and for pointing to the literature discussing the merits of the Bayesian approach in psychology and neuroscience.
>
> The papers of Wei and Stocker (Refs. [1,6]) and of Morais and Pillow (Ref. [9]) have indeed been an important inspiration for our work, which we have been careful to acknowledge. Our results, however, go beyond a simple combination of these articles. In Ref. [6], Wei and Stocker consider a model in which the encoding maximizes the mutual information and the decoding is Bayesian (in addition, they assume a property of symmetry for the likelihood function), and they show that this predicts anti-Bayesian biases, but no “Bayesian” biases. Moreover, in Ref. [1], the same authors show how this model results in a relation of proportionality (a “law”) between the bias and the derivative of the squared discrimination threshold (or of the variance); the constant of proportionality is not provided. In Ref. [9], Morais and Pillow propose a generalization of this model, with a generalized objective function that includes the mutual information but also $L_p$ reconstruction errors, and with a generalized constraint on the encoding capacity. Assuming a Gaussian likelihood, they show that the bias of the maximum a posteriori (MAP) decoder follows the “law” mentioned above. As for the Bayesian mean, they estimate it numerically in a number of simulations.
>
> We now point to our main results that are not found in the related literature. In Section 2, we derive the bias and the variance of the Bayesian mean as a function of the prior and of the encoding Fisher information, under the main assumption that a series of independent signals (not necessarily identically distributed) encodes the stimulus. Here, the encoding is not assumed to be optimized for a particular objective, and its functional form need not be specified. Under these very general conditions, we provide an analytical approximation to the bias,
> $$
> E[\hat x - x | x ] \simeq \frac{1}{I(x)}  \Bigg[ \frac{\pi' (x)}{\pi(x)} -  \frac{I' (x)}{I(x)} \Bigg],
> $$
> which is not provided in Refs. [6] and [9], and to our knowledge it is not found in the psychology and neuroscience literature. The idea that the encoding has an impact on perceptual estimates exists in the literature, including in Refs. [1,6,9]. But this is a general result that provides a clear and quantitative view, which was not previously available, of the respective contributions of the prior and of the encoding to the bias of the Bayesian mean. This clarifies what behavior one can expect from an encoding-decoding mechanism. For instance, it makes the possibility of anti-Bayesian biases readily understandable: it suffices that the precision of the encoding varies faster with $x$ than does the prior.
>
> In Section 3, we consider different objectives, including the maximization of the mutual information and the minimization of $L_p$ errors, as in Ref. [9], but also objectives that arise in discrimination tasks, which are also ecologically relevant, and not considered in Ref. [9]. We show in Section 4 how the behavior of our observer is shaped by the specifics of the task. We derive, for instance, the constant of proportionality (not previously provided) between the bias of the Bayesian mean and the derivative of its variance, and show how it depends on the objective of the task. With some tasks, e.g. the estimation task with squared-error loss, the bias is “Bayesian”, while with other tasks, e.g., the discrimination task with proportional rewards, the bias is anti-Bayesian. In comparison, the model in Ref. [6] cannot explain “Bayesian” biases: the predicted bias is always anti-Bayesian. Our theory shows how the two kinds of biases can appear in different situations. Moreover, in Ref. [6] the encoding maximizes the mutual information, which is not the objective that the decoding maximizes; by contrast, with the two tasks just mentioned (estimation task with squared-error loss and discrimination task with proportional rewards), the Bayesian mean is the optimal decoder: our model is thus consistently optimal, i.e., both encoding and decoding maximize the same objective.

---

### Official Review · Reviewer_hpiu · 2021-07-17

**Rating:** 8
**Confidence:** 3

**Summary:**

The paper derives a parsimonious theory for the performance of a Bayes mean estimator under optimal encoder-decoder model. After identifying the equations for bias and variance, the authors explain bayesian and anti-bayesian biases that depend on task, and the relationship between bias and derivative of variance.

**Ethical Concerns:**

no ethical concern

**Limitations And Societal Impact:**

No negative societal impact

**Main Review:**

It is a very well written paper. While the mathematical results might not be completely new, but as the authors say, the connections to psychophysics are novel.

* The title should be made less general, and mention the implications for perception/psychophysics.

* How is the constraint on Fisher Information $\int\sqrt{I(x)}dx \leq \sqrt{K}$ justified?
How are the mathematical results affected when you use a slightly different constraint (such as $\int I(x)^q dx \leq K^q$ for arbitrary $q$ for example)? Do the main insights (such as anti-bayesian bias for certain tasks) still hold ?

* Under what practical conditions are the assumptions of the theory violated? Listing them concisely would help a reader applying it to practical data.

* Moreover, the utility of theory would be much more apparent if a re-interpretation of existing empirical data would be presented. Improving the paper along this line would make the paper more impactful.

* Related, simulations with more practical considerations (finite n) and correlated neurons would help in highlighting the limitations of the theory for a reader. Another limitation is the assumption that decision reflects bayesian mean estimation.

**Time Spent Reviewing:**

6 hours

---

> ### Author Response · Authors · 2021-08-09
> **Response to Reviewer hpiu**
>
> We thank Reviewer hpiu for the positive comments on our work.
>
> The constraint on the Fisher information that we consider has been used in the literature, in particular in Ref. [6] for a similar optimization problem. This constraint on the possible variation in the Fisher information over the sensory space arises in any model where one assumes that information must be transmitted through a communication channel that produces a noisy output signal $r$ when an input signal $m$ is supplied, where $m$ (like the physical feature to be encoded) is unidimensional, though $r$ need not be.
>
> Suppose that the conditional probabilities $p(r|m)$ are given by the biophysics of the channel, but that both the way that physical states $x$ are encoded as inputs to the channel (i.e., the encoding function $m = m(x)$) and the way that an estimate of the physical state is produced using the output signal (i.e., the decoding function $\hat x = \hat x(r)$) are to be optimized for a given environment (prior distribution over states $x$) and decision problem. Then in the case of any differentiable, monotonically increasing encoding function $m(x),$ the Fisher information at $x$ is equal to
>                  $$
>                  I(x) = \langle (\frac{\partial \log p(r|m(x))}{\partial x})^2\rangle = \langle (\frac{\partial \log p(r|m(x))}{\partial m} \cdot m^\prime(x))^2\rangle = m^\prime(x)^2 \cdot I^*(m(x)),
>                  $$
> where $I^*(m)$ is the Fisher information of the channel at input signal $m$ (which is independent of the encoding rule). By reparameterizing the input space (which is innocuous if we allow arbitrary monotonic encoding rules), we can choose a measure of $m$ such that $I^*(m) = k^2 > 0$ over the entire input range of the channel. It then follows that $(I(x))^{1/2} = k m^\prime(x),$ so that for any monotonic encoding rule, $\int (I(x))^{1/2} dx = k (\bar m - \underline m),$ where $\underline m$ and $\bar m$ are the lower and upper bounds of the range of the encoding function. If the properties of the channel require $m$ to remain within a bounded range, then this range (called the "Thurstone invariant" by Zhang, Ren and Maloney, 2020) determines a finite upper bound for $\int (I(x))^{1/2} dx$.
>
> As a concrete example of such a coding problem, we assume in our calculations that the internal representation $r$ is a real number, drawn from a Gaussian distribution $N(m(x), \nu^2)$, where $\nu$ is independent of the input $m(x)$. This is an example of a communication channel for which the Fisher information is uniform over the channel input space, $I^*(m) = \nu^{-2}$ for all $m$. As a more biophysically realistic example, Ganguli and Simoncelli (2010) propose a model with a heterogeneous population of neurons whose tuning curves "tile" the line corresponding to different values of $m$, a nonlinear transformation of the stimulus space (where the nonlinear transformation is to be optimized). In this example, $r$ is high-dimensional (it corresponds to a vector of spike counts by the different neurons in the population), but the probabilities $p(r|m)$ are taken as given, and moreover satisfy a translational invariance that imply (as in our simpler model) that $I^*(m)$ is uniform over the input space $m$. The finiteness of the range of possible values of $m$ for which this uniform Fisher information can be maintained then follows from the finiteness of the population of neurons (Ganguli and Simoncelli link the Fisher information of their channel to the local density of the population of neurons in each range of values of $m$). Ref. [7] derives the same form of resource constraint in a model of a neuron whose firing rate follows a sigmoidal tuning curve; here the input signal $m$ corresponds to a monotonic transformation of the neuron's firing rate (so that the encoding function $m(x)$ is effectively the neuron's tuning curve), while the output signal $r$ is the number of spikes produced. In this case, the bounded range for $m$ reflects a bounded range of possible firing rates for the neuron.
>
> In addition, Wei and Stocker (2016) argue that the quantity bounded in this constraint is proportional to the number of different stimuli that the encoding is able to discriminate, thus providing a measure of the encoding capacity of a neural system. They note, moreover, that this formulation of the resource constraint has the attractive feature of  being invariant under reparameterization of $x$.
>
> In spite of these examples, we agree with the Reviewer that different constraints might be relevant, such as that suggested by the Reviewer, and considered in Ref. [9]. For the sake of concision (and because of the page limit), we have not developed, in the article, our results with this kind of constraints, but it is straightforward to do so. With the generalized constraint
> $$
> \int I(x)^{q/2} dx \leq K^{q/2},
> $$
> we find that the optimal Fisher information is proportional to the prior raised to the power $\Gamma=2a/(p+q)$, with $a$ and $p$ depending on the task, and defined as in the paper. It follows that many results in the article are the same, in this case, if we replace $\gamma$ with $\Gamma$; in particular, regarding the variance and the bias:
> $$
> v(x) \propto \frac{1}{\pi(x)^\Gamma}
> $$
> and
> $$
> b(x) = \frac{\Gamma-1}{\Gamma} v'(x).
> $$
> If $q=1$, then the constraint is the same as in the paper, and $\Gamma=\gamma$, thus we obtain the same results. The constant $\Gamma$ determines the strength of the anti-Bayesian effect, e.g., if $\Gamma>1$, the overall bias is anti-Bayesian. A larger $q$ results in a smaller $\Gamma$, and thus in less tasks in which anti-Bayesian biases should appear, and more tasks with “Bayesian” biases; and conversely. For instance, with $q=2$, which corresponds to a bound on the integral $\int I(x) dx$, the “discrimination task with proportional rewards” yields no bias ($b(x)=0$), while when $q=1$ it yields an anti-Bayesian bias (Table 1). Conversely, in the extreme case $q \to 0$, all the tasks considered in the article result in an anti-Bayesian bias, except the estimation task with squared errors, which results in no bias (while it yields a “Bayesian” bias with $q=1$).
>
> As for the assumptions of the theory, they are as follows: the encoding consists in an accumulation of independent signals; it is optimized for the given task, under the constraint mentioned above; the decoder is the Bayesian mean (which is optimal for two of the tasks we consider); and the noise is small. Note that the assumption of optimal encoding and decoding implicitly entails that the decision-maker knows the prior. Regarding the noise, the simulations we run are intended to provide some sense of the impact that the degree of noise has on the results, in comparison with the theory. Violations of the other assumptions would distort the results in ways that would be interesting to explore, but that are outside the scope of this paper. As for correlated neurons, our derivations rely on a central limit theorem, and we surmise that they should withstand some amount of correlation in the encoding; we agree with the Reviewer that this would be an interesting direction to investigate.
>
> Finally, we agree that empirical data would substantiate our theoretical results. One prediction of the theory, however, is that the bias should depend on the objective of the task, but it is not common in psychophysics experiments to have subjects run two similar tasks that differ by their objectives. In many experiments, moreover, the objective and the prior are not precisely defined. Nevertheless, we comment in the paper on the results of Brayanov and Smith (Ref. [17]), who run two experiments. In the first experiment, subjects are asked to hold one hand steady while an object is removed from it: this implicitly requires subjects to estimate the weight of the object. The second experiment is a discrimination task between two weights. In the former experiment they find that biases are directed towards the prior expectations, while in the latter experiment they find biases directed away from the prior expectations, which is consistent with our predictions regarding estimation and discrimination tasks. In order to provide further empirical results, we are currently conducting experiments that are designed to test various predictions of the theory.
>
> Ganguli, D., and Simoncelli, E. P. (2010). Implicit encoding of prior probabilities in optimal neural populations. In J. D. Lafferty, C. K. I. Williams, J. Shawe-Taylor, R. S. Zemel, \& A. Culotta (Eds.), Advances in Neural Information Processing Systems 23 (pp. 658--666).
>
> Wei, X.-X., and Stocker, A. A. (2016). Mutual Information, Fisher Information, and Efficient Coding. Neural Computation, 326, 305–326. https://doi.org/10.1162/NECO_a_00804
>
> Zhang, H., Ren, X., and Maloney, L. T. (2020). The bounded rationality of probability distortion. Proceedings of the National Academy of Sciences, 117 (36): 22024-22034.

---

### Official Review · Reviewer_5yDC · 2021-07-18

**Rating:** 7
**Confidence:** 3

**Summary:**

The authors present a unifying account of how the bias and variance of the Bayesian mean estimator (in an encoder-decoder setup) are determined from the prior and the encoder’s Fischer information. They then use these approximations to the bias and variance to explore the behavior of Bayesian-optimal systems across a fairly wide range of objectives (estimation, MI maximization, discrimination). Their theoretical findings recapitulate known findings, and also offer an explanation for anti-Bayesian bias (when the bias is toward the stimulus with lower prior probability). Specifically, the bias is proportional to the sum of a Bayesian effect (toward the prior) and an anti-Bayesian effect, where the dominant direction depends on the specifics of the problem. The authors show that their approximations are fairly consistent with true values for a simple problem when the encoding noise is low.


**Ethical Concerns:**

None.

**Limitations And Societal Impact:**

The discussion of limitations is adequate.


**Main Review:**

Overall, this paper is fairly well-written, clear, and well-thought out. The theoretical work appears sound, the simulations are fairly convincing, and the conclusions seem interesting. (However, being somewhat outside this exact field, I hope to leave it to others to assess the novelty of this work vis-a-vis other works that have promised to unify Bayesian and anti-Bayesian biases under a single framework.)

That said, one thing that would have made this paper more impactful to me would have been a more thorough interrogation of _why_ Bayesian or anti-Bayesian biases are optimal in specific situations. It seems like this is a deep question into which these authors have been granted a privileged viewpoint—their approximation to the bias. More specifically, I wanted the authors to spend more time unpacking the I’(x)/I(x) term, what it means, and what intuition we might gain that would help us guess which problems will have Bayesian or anti-Bayesian bias.


**Time Spent Reviewing:**

2

---

> ### Author Response · Authors · 2021-08-09
> **Intuitive explanations on how the task determines the direction of the bias**
>
> We thank Reviewer 5yDC for her/his positive comments on our work. We agree that providing more intuitive explanations of our results, and of which problems one should expect to result in a Bayesian or an anti-Bayesian bias, would be helpful to the reader.
>
> The anti-Bayesian component, as noted by the Reviewer, depends on the variations of the Fisher information, $I(x)$: larger variations result in a stronger anti-Bayesian component, that may be greater than the “Bayesian” component. The Fisher information determines the precision with which a given stimulus, $x$, is estimated; to what extent the precision matters depends on the task at hand. We consider, for example, an estimation task with a loss function equal to $|\hat x-x|^p$, where $\hat x$ is the estimate (as discussed in the article, the expected loss in this case is approximately the average of $1/I(x)^{p/2}$). If $p$ is large, then small errors do not matter greatly; while if $p$ is small, the smallest error in the estimate results in a large loss. Thus with small $p$, the precision with which a stimulus is observed will have a large impact on the task performance; and given a limited “budget” of precision capacity, it will be beneficial to allocate more precision to the more frequently occurring stimuli (e.g., $I(x) \propto \pi^2 (x)$, in the limit $p\to0$). By contrast, with large $p$, a small estimation error will not yield a large loss, even for a stimulus that occurs frequently. This results in a different optimal allocation of the precision, with a “flatter” Fisher information (in the limit $p \to \infty$, $I(x)$ is constant).
>
> In other words, the variations of the optimal Fisher information — and thus the strength of the anti-Bayesian component — depend on the extent to which small errors are costly, in the task at hand. This also applies to other tasks, such as the discrimination tasks examined in the paper: with a “discrimination task with constant rewards”, mis-ordering two stimuli that are extremely close may result in the maximum possible loss; while with a “discrimination task with proportional rewards”, it would only result in a loss proportionally as small as the difference between the two stimuli. Consequently, the former task results in an optimal Fisher information that varies more strongly with the prior than that in the latter task ($I(x) \propto \pi^2 (x)$ vs. $I(x) \propto \pi^{4/3}(x)$; see Table 1). As for the objective of maximizing the mutual information (last line in Table 1), it is equivalent to the limiting case $p \to 0$ of the estimation task described above, i.e., to a task in which all mis-estimations of the stimulus are equally costly, no matter how small the distance is between the stimulus and the estimate.
>
> Finally, the anti-Bayesian component of the overall bias, $-I'(x)/I(x)$, is directed towards stimuli with decreasing precision. Indeed, a stimulus that is less precisely-encoded yields imprecise representations that a more precisely-encoded stimulus would be comparatively unlikely to yield; as a result, the estimated stimulus, decoded from these representations, tends to be “pushed away” from the more precisely-encoded stimulus, and towards the less precisely-encoded stimulus.
>
> In summary, whether or not a given task is susceptible to result in anti-Bayesian biases depends on the cost of errors in the task. If the loss (or reward) function is very sensitive to small errors, then the bias tends to be anti-Bayesian. If small errors result in small losses, then the bias tends to be “Bayesian”, i.e., in the direction of the more probable stimuli.

---

### Decision · Program_Chairs · 2021-09-27

**Decision:**

Accept (Spotlight)

**Comment:**

This is a very clearly written paper that introduces a "parsimonious theory for the performance of a Bayes mean estimator under optimal encoder-decoder model. After identifying the equations for bias and variance, the authors explain bayesian and anti-bayesian biases that depend on task, and the relationship between bias and derivative of variance." (Reviewer hpiu)

The main results are derived combining existing work, but the connections to psychophysics are novel. The findings (for example the emergence of contradictory biases under a consistently-optimal encoding-decoding model, that depend on the cost of errors in the task) are very interesting. Supported also by the highly positive sentiment of the reviews, I anticipate that the paper will be perceived with interest by a broad audience, hence I suggest a spotlight.